# Unveiling the "Veil" of information disclosure: Sustainability reporting "greenwashing" and "shared value"

**Wei Xu**[1☯], **Mingzhu Li**[1☯*], **Sen Xu**[2]

1 Business School, Xi'an University of Finance and Economics, Xi'an, Shaanxi, China, 2 Management School, Xi'an University of Finance and Economics, Xi'an, Shaanxi, China

☯ These authors contributed equally to this work.
* 2121031016@xaufe.edu.cn

**Data Availability Statement:** The data used in this study are third-party data owned by CSMAR (China Stock Market & Accounting Research Database), therefore, the authors have no right to share the

## Abstract

With the increasing attention of the capital market to environmental, social and governance information, sustainability reporting has become an important carrier for stakeholders to gain insight into sustainability of companies. But the emerged "greenwashing" problem has also brought haze to the value creation of capital market. To study the consequences of the pseudo-social responsibility behavior of "greenwashing", this paper takes China's listed companies as the research sample to empirically examine the relationship between sustainability reporting "greenwashing" and "shared value" creation. It is found that the "greenwashing" behavior of corporate sustainability reporting significantly reduces the "shared value" creation, while the degree of sustainability information asymmetry and the quality of information disclosure play a partial mediation role between them. Further analysis shows that the more effective internal control of a company and the greater pressure of external media supervision, the more conducive to weaken the negative impact of "greenwashing" on "shared value" creation. This paper enriches the literature on the economic consequences of "greenwashing" in sustainability disclosure and the influencing factors of "shared value" creation, extends the research on information disclosure and "shared value" from financial information to non-financial information. The results call for the state to promote legislative work, formulate unified standards and compress the "greenwashing" gray space; Governments could implement mandatory disclosure, implement independent authentication and strengthen "greenwashing" social supervision; Companies should strengthen capacity building and improve the "greenwashing" governance mechanism with the help of digital empowerment.

## Introduction

Adopted by United Nations Member States in 2015, the 2030 Agenda for Sustainable Development defines the world's vision and priorities for sustainable development for human, planet, prosperity, peace and partnership by 2030, and is committed to mobilizing global efforts to achieve the 17 Sustainable Development Goals (SDGs) [1]. Many countries have plans in place

data. Interested persons can contact CSMAR for the data (see https://www.gtarsc.com/ for more details, contact via email: service@csmar.com, tel: 400-639-8883). The dataset is in "Corporate Research Series". The authors confirm that they did not have any special access or privileges to the data that other researchers would not have.

**Funding:** The study was funded through the National Statistical Science Research Program Project (2021LY081) and Social Science Foundation of Shaanxi Province(2022HZ1516). The funders had no role in study design, data collection and analysis, decision to publish, or preparation of the manuscript.

**Competing interests:** The authors have declared that no competing interests exist.

to implement the SDGs. In 2016, China organically integrated the Sustainable Development Agenda with its medium- and long-term development plan, and incorporated the SDGs into its 14th Five-Year Plan in 2021. Companies are key stakeholders in achieving the SDGs. With the development of the social responsibility movement and the increasing attention of the capital market to corporate sustainability, the importance of sustainability reporting has become increasingly prominent. According to KPMG's 2020 Sustainability Report Survey, 80% of the world's 100 largest companies report their sustainability [2], and the "responsibility" of reporting has begun to change to the "sustainability" side, which has led to a transition of social responsibility reporting to sustainability reporting. And the sustainability information disclosure has become an important window for stakeholders to obtain non-financial information of companies and realize the externalities of business activities. Under the "Carbon Peaking and Carbon Neutrality" goals, it is crucial to explore the green governance effect and mechanism of sustainable development, effectively improve the quality of information disclosure in sustainability reporting, and finally achieve green and sustainable development in China.

According to the new requirements for information disclosure put forward by economic development in the new era, the concept of sustainability requires companies to shift from traditional values to the "shared values", which ensure the coordinated development of economic, social and environmental values [3]. Value creation is the original intention and mission of a company, while sustainability is the vision and goal. In the face of increasingly stringent environmental regulatory requirements and public environmental protection needs, companies may fully disclose relevant practice information in accordance with the requirements of sustainability reporting, and actively innovate and develop sustainable businesses to highlight their awareness of sustainability [4]. However, in fact, some companies use the market's preference for sustainability reporting to exaggerate their sustainable advantages in management or business, through "greenwashing" behavior to make investors and other stakeholders accept their social and ecological value, ability or contribution. As shown in Fig 1, the "China

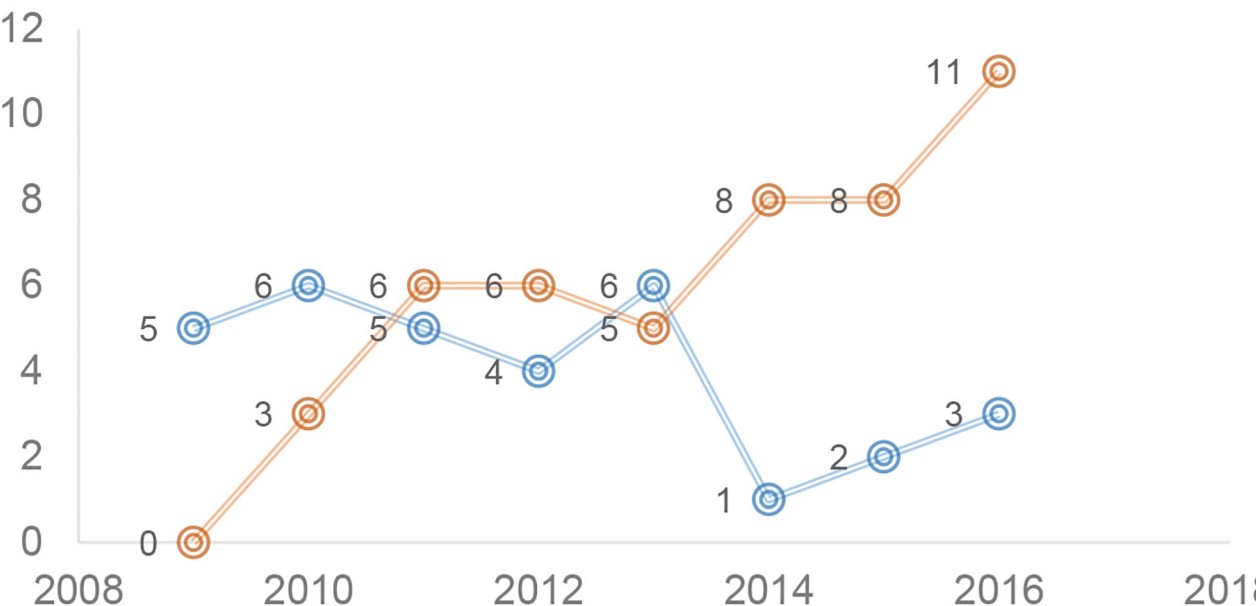

**Fig 1. The number of corporations selected in the "China Greenwashing List" from 2009–2016.**

Greenwashing List" continuously issued by the mainstream media Southern Weekend, and typical events reported by financial media——Coca-Cola Company "greenwashing" incident, PetroChina "greenwashing" incident, Volkswagen emissions scandal, etc., are not uncommon.

The "greenwashing" in sustainability reporting has characteristics and complexity and confidentiality, which may have different degrees of economic consequences for various stakeholders of the company, thereby hindering the achievement of "shared value" goals [5]. This research attempts to answer the following questions: Does the "greenwashing" behavior of corporate sustainability reporting and its information disclosure affect the "shared value" creation of enterprises? If there is an influence between them, how does the influence path unfold? and What kind of constraint mechanism can maximize the negative impact of the "greenwashing" behavior in corporate sustainability reporting?

Accordingly, this paper comprehensively considers the "shared value" creation needs of environmental, social, enterprise and other stakeholders, and use the data of China's listed companies for empirical testing. It is found that the "greenwashing" behavior of information disclosure in corporate sustainability reporting significantly reduces "shared value" creation, and the degree of sustainability information asymmetry and information disclosure quality play a partial role as an intermediary between them. Further analysis shows that the more sound the internal control of the enterprise and the greater the pressure of external media supervision, the more conducive it is to weaken the negative impact of "greenwashing" on "shared value" creation. The research conclusion enriches the relevant literature on the economic consequences of "greenwashing" in sustainability disclosure and the influencing factors of "shared value" creation, and extends the research of information disclosure and "shared value" from financial information to non-financial information, which provides scientific evidence for improving the quality of information disclosure in sustainability reporting, achieving the goal of "shared value" creation, and further improving corporate sustainable development capabilities.

The next section presents the literature review and then is research method. The research finding and discussion are presented in the subsequent section, and, finally, the conclusions are expressed.

## Literature review

"Shared value" was first defined as policies and business practices that enhance a company's competitiveness and improve the conditions of its communities and societies [6]. It was mainly presented in the social value reflected in the increase of income and environmental improvement, commercial value reflected in the increase of profit and competitive position improvement [7]. To measure these values, ten linked modules including strategy, assets and key partners were used as metrics [8], but because the field was still in its early stages, few companies could combine these modules to complete the entire process. Subsequently, specific financial, asset and other indicators were used to measure [9]. In the 21st century, more and more companies are disclosing sustainability reporting that reflect their economic, social and environmental performance, also known as the "triple bottom line" of a company, as shown in Fig 2.

Its theoretical origin could be traced back to the "triple bottom line" theory created by British sustainability consultant John Elkington in the 90s of the 20th century. Economic value refers to the healthy operation of enterprises, pay taxes, and create more value for shareholders and society; Social value refers to the harmonious coexistence between enterprises and stakeholders, including employees, suppliers, customers, governments and the public; Environmental value refers to the optimal use of natural resources, the protection of environmental resources, the production of green products, etc. [10]. As shown in Fig 3. The "triple bottom

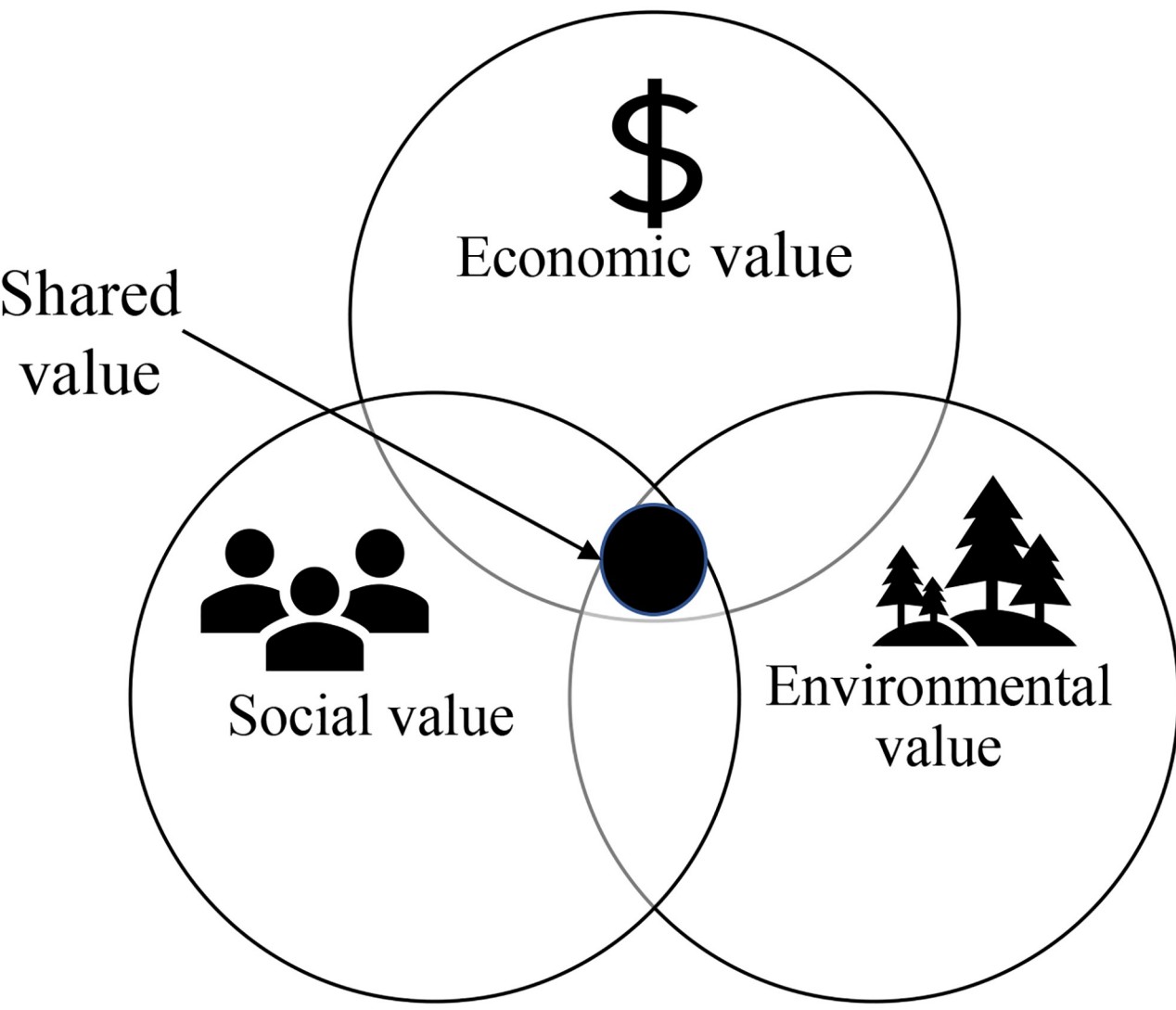

**Fig 2. "Shared value" that balances economic, social, and environmental values.**

line" theory holds that a company that contributes to social progress by creating economic, social, and environmental value at the same time is a real sustainable company [11, 12]. Therefore, this paper uses this perspective to comprehensively measure the "shared value" creation activities of companies from three aspects: economic value, social value, and environmental value.

Creating "shared value" is the fundamental purpose of a company's existence [13], so it is important to explore the factors that affect "shared value" creation. Relevant research is mainly divided into two aspects: on the one hand, there are negative influencing factors, insufficient innovation ability of suppliers leads to negative information sharing value [14]; Misinformation between retailers and manufacturers can also have a negative impact on creating "shared value" [15]. On the other hand, there are positive influencing factors, the use of big data will promote the creation of "shared value" [16]; The public's perception of high-quality e-government content and cross-departmental assistance from Internet platform enterprises also have a positive impact on "shared value" [17, 18].

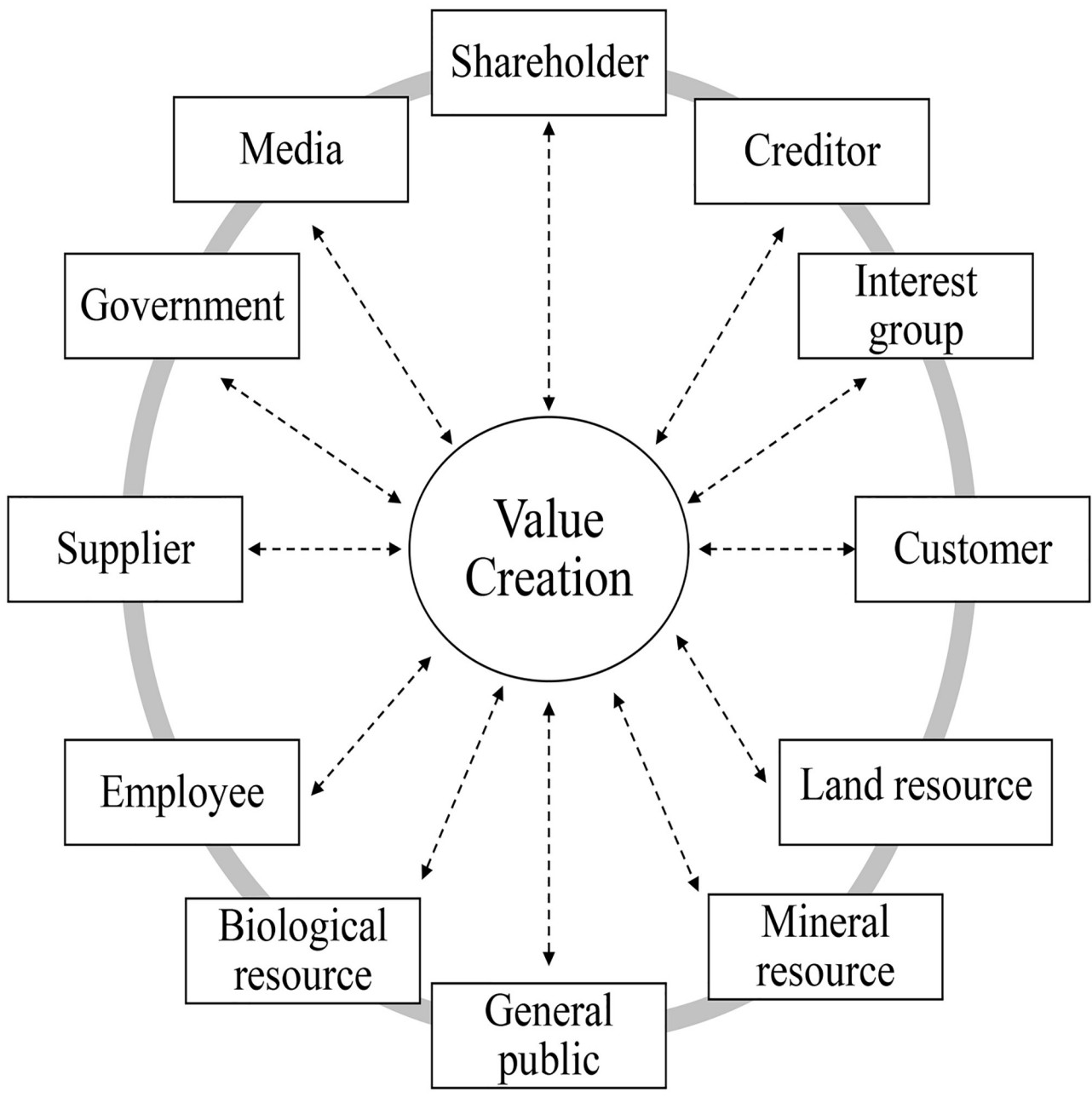

**Fig 3. Interaction between stakeholders in value creation.**

The so-called "greenwashing" mainly refers to the false publicity, exaggeration or even misleading of various green environmental protection behaviors in the daily production, operation and management of companies. While in fact they do not really take green environmental protection measures, or even go the other way, which create a good environmental protection enterprise image by glossing over their own environmental pollution and resource waste behaviors [19, 20]. The "greenwashing" of sustainability information not only includes the behaviors in the fields of production and operation, product quality and management and marketing, but also involves the behaviors of whitewashing and covering up a series of areas

related to the sustainability of one company, such as abiding by business ethics, supporting charity, protecting the rights and interests of employees, and donating to social welfare.

The driving factors of "greenwashing" behavior are mainly reflected in three aspects. First, from the perspective of marketing, companies that flaunt green products can get extensive attention from the public, which is conducive to improve their profit [21]. And compared with real green products, "greenwashing" products have low cost and high price, so companies have greater profit space [22]. Second, from the perspective of corporate management, the internal organizational structure of a large company is complex and the decision-making is scattered, therefore the top management may ignore the "greenwashing" behavior for the sake of benefit distribution [23]. Finally, from the reputation perspective, "greenwashing" is considered a symbolic response to social responsibility and a reputation strategy [24].

Regarding the measurement of the degree of "greenwashing", a unified standard has not yet been formed, and the quantitative research on the degree of "greenwashing" of companies in academia mainly focuses on two aspects. First, identifying the location where the "greenwashing" information appears, using "deep greenwashing" to explain [25]. Or directly carrying out the qualitative stratification of the degree by dividing the degree of "greenwashing" into "hard greenwashing" and "light greenwashing" to measure [26]. Second, constructing an indicator system from four perspectives, including governance and structure, process and control, and characterizing them through the degree of selective disclosure and manipulation disclosure [27]. This paper argues that this multi-dimensional "greenwashing" degree indicator system is more objective and reasonable [24].

At present, the research data on the impact of "greenwashing" behavior on "shared value" is scarce, but "shared value" is actually a comprehensive concept, which can be reflected in the financial performance, environmental performance, and social behavior of companies to a certain extent. First, fulfilling social responsibility can help companies profit from capital markets, and in the long run, substantial environmental actions can improve corporate financial performance [28]; Once the "greenwashing" activity is recognized by investors, the capital market will respond to it [29], which is reflected in the more serious the "greenwashing" situation, the lower the cumulative excess yield during the window period. Secondly, "greenwashing" activities will reduce the enthusiasm of companies for green product research and development, reduce consumers' willingness to consume green products [30], mislead stakeholders to make decisions, and lead to a vicious circle of corporate environmental performance. Finally, "greenwashing" will imitate and spread within the group, forming a "ripple effect", causing other companies to imitate [31], distorting the ethical standards of the industry or region, and people will question the authenticity of the proposition of social responsibility, and eventually weaken the social responsibility awareness of the entire group.

The phenomenon of "greenwashing" is nothing new now, and it is an emergency to control the problem of "greenwashing". But the governance of "greenwashing" problem first requires the guarantees of relevant systems and laws, and the information disclosure standards also need to be more detailed and definite [24, 32]. Meanwhile, it is obviously efficient that stakeholders such as government, companies, and the public coordinate to control the "greenwashing" behavior [33]. Therefore, further improving the sustainability reporting information disclosure system, preventing corporate violations, and proposing an effective "filtering" governance strategy framework can provide significant support for the green and sustainable development of companies and the high-quality development of the country.

In summary, the existing research provides theoretical guidance and thoughtful reference for the study of this paper. However, the following issues need further research: First, the influencing factors of "shared value" creation currently mainly focus on the research of various factors such as innovation ability and big data on "shared value" creation, and the exploration of

situational influencing factors such as "greenwashing" is insufficient and scattered. Second, since the information effect is subject to the quantity and quality of information disclosure, this paper distinguishes the information effect brought by the "greenwashing" behavior into the degree of information asymmetry (quantity) and the quality of information disclosure (quality), and explores the mediating role of information asymmetry and information disclosure quality respectively. Third, existing research mainly studies the governance of "greenwashing" behavior from the perspectives of policy system and macro collaborative governance, however, there are few studies on multi-level and integrated governance strategies that "filter" sustainability reporting. According to the accurate identification of "greenwashing" behavior of corporate sustainability information, this paper's findings reveal the mechanism of "greenwashing" behavior and a company's "shared value" creation. It also explores scientific and reasonable "filtering" governance strategies, which can extend the research on information disclosure and "shared value" from financial information to non-financial information. The findings could help to improve the quality of information disclosure in sustainable development reports and encourage companies to achieve the goal of "shared value" creation, and further enhance sustainable development capabilities.

## Research assumptions

With the boom of sustainable development theory, companies should uphold the principle of "shared value" creation, improve their awareness of social responsibility and environmental protection, and create social and environmental value while producing economic value. Sustainability reporting provides an effective way for shareholders and other related to supervise corporate responsibility. In addition, improving the level of sustainability information disclosure is also one of the efficient ways to enhance "shared value". However, due to the natural opportunistic tendency, many companies may occur "greenwashing" behaviors in the process of disclosing sustainability information, which not only weakens the quality of information disclosure in sustainability reporting, but also causes information asymmetry. It could harm the legitimate rights and interests of stakeholders, and then affect the creation of "shared value". The logical framework of this research is shown in Fig 4.

### Sustainability reporting "greenwashing" and "shared value"

Compared with the financial information disclosed by companies, the scope of sustainability information covers not only their economic performance, but also performance in the process of fulfilling its social responsibilities such as protecting the environment and protecting the rights and interests of employees. Therefore, sustainability mainly includes economic, environmental and social triple performance information. Companies in addition to pursuing their own profit maximization, abide by industry standards, professional ethics, laws and regulations, fulfilling environmental protection, safety, contribution and other responsibilities, true and completing disclosure of relevant sustainability information has become a general consensus of society. At present, in terms of information disclosure in corporate sustainability reporting, the negative effects of "greenwashing" behavior are often more serious than the simple lack of corporate environment and social responsibility, and deception are more misleading and harmful to stakeholders.

For the company itself, from the perspective of direct impact, "greenwashing" behavior will increase two major costs. One is the cost of "dissembling": "greenwashing" is only a false statement after all, in order to smoothly practice deception, the company is bound to pay a certain cost of manpower and material resources. And with the increasing of external supervision and investor attention intensity, the cost will also rise. The other is the cost of punishment: truth

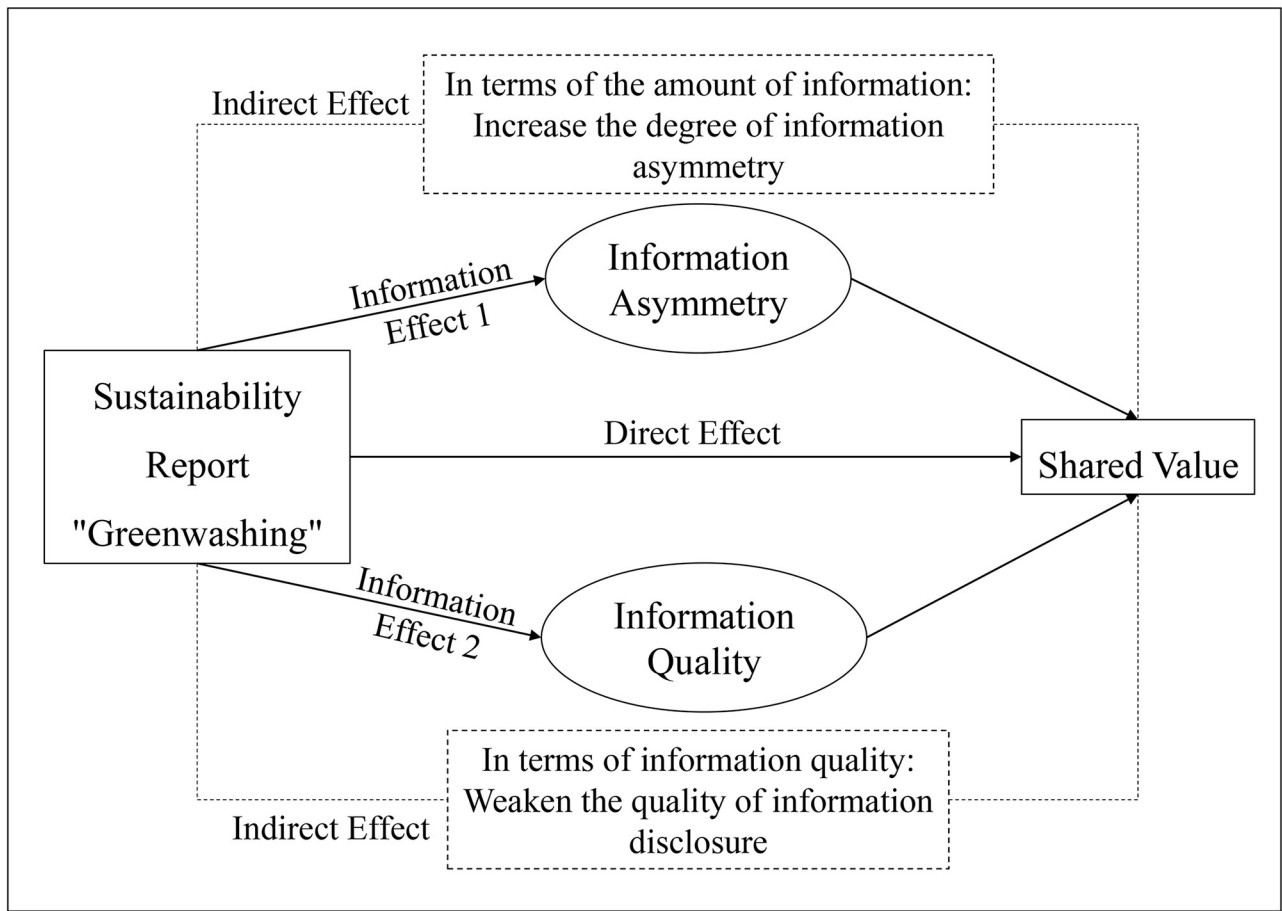

**Fig 4. The research framework of this article.**

will come to light sooner or later, companies that do not really accomplish energy conservation and emission reduction or even excessive emissions to pollute the environment will be punished with certain penalty fees. And with the improvement of the relevant legal system and the increasing level of punishment, the fee will also rise.

From the perspective of indirect impact, "greenwashing" will also affect the company's own benefits for a long time, and this negative impact is irretrievable by the value-added of corporate benefits created by improving the quality of sustainability reporting disclosure. First, companies are likely to become dependent after tasting the sweetness, so as to settle for the status quo and stagnate their emission reduction actions, which may increase the penalty costs due to excess emissions, or may reduce government rewards or subsidies, thereby reducing corporate efficiency. Second, the "greenwashing" behavior will affect the moral perception of employees within one company, reduce employees' sense of belonging and loyalty to it, damage the innovation ability and work enthusiasm of employees, and negatively affect the value added of corporate benefits in the end [34]. Third, once the "greenwashing" behavior is identified, the capital market will respond immediately. It will lead to the withdrawal of capital due to the loss of investors' trust even if the high-quality sustainability information is disclosed at this time, which will seriously affect the value of the company's stock. This is very detrimental to the promotion of "shared value", instead the benefits outweigh the losses [35].

The theory of legitimacy holds that a company's level of legitimacy is one of the ways to express the degree of fulfillment of the social contract, and any company must conform to social values and achieve legal compliance in various business activities. However, companies naturally have opportunistic tendencies, there is a complex principal-agent relationship between companies and various stakeholders. While "greenwashing" behavior adopts different words and deeds of operation and management, which will make companies lack energy saving, carbon reduction, efficiency increase brought by "true green". Through false publicity of green, cover up pollution evil and other production and operation management methods that run counter to the concept of environmental protection, it seriously infringes on the interests of investors, managers, consumers, suppliers, customers, potential investors, governments and other stakeholders. It also affects the basic living environment of the public and even the region. The unfair competition caused by "greenwashing" behavior has caused production capacity imbalance, resource bottlenecks, ecological risks, natural disasters, etc., which have also brought unpredictable harm and losses to the entire market economy and society, and the vicious chain reaction after the exposure of "greenwashing" behavior of enterprises will also affect public health, sustainability of the industry and the rational allocation of resources in the whole society. To this end, government regulatory departments and environmental protection organizations have to intervene and pour the human, material and financial resources that were originally committed to sustainable development into "greenwashing and anti-counterfeiting" to eliminate the prevalence of pseudo-social responsibility and eventually cause the overall welfare of society and environmental performance to decline. In view of this, this paper proposes the following assumptions:

**Hypothesis 1**. The "greenwashing" behavior of the sustainability reporting information disclosure has a negative correlation with "shared value" creation.

## Information effects, sustainability reporting "greenwashing" and "shared value"

Under the framework of information economics, information is the basis and premise to ensure the accuracy of decision-making, the information is more accurate, the more guaranteed the decisions made, so information resources play an important role in the capital market. And corporate "greenwashing" sends the wrong sustainability signal to external stakeholders. Consider that the effect of information is subject to both quantitative and qualitative factors of information disclosure: On the one hand, when external stakeholders obtain less information about the enterprise, it indicates that there is a serious information asymmetry between the supply and demand sides of the market; On the other hand, when external stakeholders obtain certain information, the accuracy and validity of the information are equally important. In further in-depth discussion of the possible impact paths, this paper explores the mediating effects of information asymmetry and information disclosure quality by distinguishing between information asymmetry degree (quantity) and information disclosure quality (quality) respectively.

Sustainability reporting, as a positive and well-known signal, is directly related to the real performance of a company. That is to say, when the company's internal information disclosed from sustainability reporting can be generally accepted, there must be a sufficient amount of information behind it to disclose and be identified by the outside, which can reduce the degree of information asymmetry of investors [36]. However, the green market has typical information asymmetry characteristics. According to the information asymmetry theory, most of the external investors, consumers and other stakeholders have defects in the cognition of green products and green behavior information, and generally speaking, they can only indirectly

know the products from advertising and other means [37]. Thus, companies are the advantage of information which master a variety of internal and external information, while stakeholders are the disadvantage of information and the cognition of companies mainly comes from the information obtained and their own experience [38]. It is difficult to judge whether the products or services provided by the company are "really green" or "greenwashing", which brings about a significant reduction in the transparency of sustainability information and also gives companies loopholes to take "greenwashing" behavior to seek their own interests, but at the expense of the interests of other stakeholders. This is a serious violation of the principle of fairness. Consequently, it may reduce the willingness of other stakeholders to cooperate with the company, even the sustainable development capacity of the company. In the long run, "greenwashing" behavior undermines rather than enhances the value of the stakeholders, and is not conducive to the creation of "shared value". In view of this, this paper proposes the following assumptions:

**Hypothesis 2**. The degree of information asymmetry plays a mediation role between the "greenwashing" behavior of the sustainability reporting information disclosure and "shared value" creation.

With the continuous development of digital information and network technology, as well as the enhancement of public's awareness of environmental protection and comprehensive quality, stakeholders have gradually improved their ability to identify and measure the disclosed information. They hope to grasp more adequate qualitative and quantitative information of companies. For example, whether the information disclosure is accurate will directly affect the success or failure of investors' investment, and when credit rating agencies generally have high ratings on bonds, they cannot play the function of risk warning when there is a risk of default, which in turn sends wrong signals to investors. The same is true for the sustainability reporting information disclosure. After all, "greenwashing" behavior is a manifestation of false statements, and companies also understand the serious consequences caused by "greenwashing" behavior, so they will be careful every minute. The most typical example is that the company has announced a very exquisite social responsibility reporting or sustainability reporting, but has not released specific and reasonable quantitative information, resulting in an impression of futility. Moreover, companies which have "greenwashing" behavior dare not tell the truth. The more obvious this secretive performance, the lower the quality of sustainability information, the lower the effectiveness of information for stakeholders, and the more detrimental to creating "shared value". In view of this, this paper proposes the following assumptions:

**Hypothesis 3**. The quality of information disclosure plays a mediation role between the "greenwashing" behavior of the sustainability reporting information disclosure and "shared value" creation.

## Methodology and data

### Model construction

For Hypothesis 1, the following regression model is constructed to test the impact of "greenwashing" of the sustainability reporting on "shared value":

$$\text{CSV}_{it} = \alpha_0 + \alpha_1 \text{GI}_{it} + \alpha_2 \text{CONTROLS}_{it} + \alpha_3 \text{INDUSTRY} + \alpha_4 \text{PROVINCE} + \varepsilon_{it} \tag{1}$$

For Hypothesis 2–3, reference to the research of Wen et al. [39], the following regression model is established to test the mediation role of information asymmetry and information

disclosure quality between the "greenwashing" of sustainability reporting and "shared value":

$$M_{it} = \beta_0 + \beta_1 GI_{it} + \beta_2 CONTROLS_{it} + \beta_3 INDUSTRY + \beta_4 PROVINCE + \varepsilon_{it} \quad (2)$$

$$CSV_{it} = \gamma_0 + \gamma_1 GI_{it} + \gamma_2 M_{it} + \gamma_3 CONTROLS_{it} + \gamma_4 INDUSTRY + \gamma_5 PROVINCE + \varepsilon_{it} \quad (3)$$

In the above model, $\alpha_0$, $\beta_0$ and $\gamma_0$ are intercept items, $\alpha_1 \sim \alpha_4$, $\beta_1 \sim \beta_4$, $\beta_1 \sim \beta_4$ are coefficients, $CSV_{it}$ is "shared value", $GI_{it}$ is the "greenwashing" behavior of sustainability reporting information disclosure, $M_{it}$ is mediation variables, $CONTROLS_{it}$ are all control variables in this paper, INDUSTRY and PROVINCE are industry and province fixed effects, respectively, $\varepsilon_{it}$ is random disturbance items.

## Variable design

The dependent variable in this paper is "shared value". From the above, it can be seen that the sustainable development theory advocates the concept of creating "shared value", and companies will help promote the value sharing with society and the environment by investing in social and environmental improvement activities that can enhance their competitiveness, and form a symbiotic relationship in which corporate success and social progress complement each other. In general, corporate "shared value" creation activities can create different types of value for stakeholders, namely economic, social and environmental [40].

For the measurement of economic value ($X_1$), this paper refers to the research of He et al. [41], and uses Tobin's Q which reflects market indicators to measure economic value.

For the measurement of social value ($X_2$), this paper designs the indicator system of social performance from four aspects: government, employees, suppliers and consumers. Reference to the measurement of the contribution of companies to the government, employees, suppliers and consumers by Mao et al. [42], we design a measurement indicator system of social value, and take the average value of the calculation results of each indicator as the measurement indicator of social performance. The calculation formula of each indicator is shown in Table 1.

For the measurement of environmental value ($X_3$), this paper refers to Patten [43] research and uses the natural logarithm of the company's environmental capital expenditure plus 1 in the current year as the measurement of environmental value.

Finally, the entropy weight method is used to calculate the indicator weights of economic value, social value and environmental value, and the "shared value" (CSV) is calculated then. The final calculation result is shown below.

$$CSV = 0.0009865X_1 + 0.8596429X_2 + 0.1393706X_3 \quad (4)$$

The independent variable in this paper is the "greenwashing" behavior of information disclosure in the sustainability reporting. Referring to the previous research [4, 10, 44], based on the "Sustainability Reporting Guide" issued by the Global Reporting Initiative (GRI), this paper uses the content analysis method to design "greenwashing" indicator system from three

**Table 1. Social value measurement calculation formula.**

| Social Value | Government | (Taxes paid—tax refund) / main business income |
|---|---|---|
| | Employee | Cash paid to and for employees /main business income |
| | Supplier | Cost of main operations/closing balance of accounts payable |
| | Consumer | (Main business income at the end of the current period—main business income at the beginning of the current period) / Main business income at the beginning of the current period |

**Table 2. Sustainability reporting "greenwashing" indicator system.**

| Category | Indicator | Symbolic disclosure | Substantive disclosure |
|---|---|---|---|
| Readability | Report form | Non-disclosure | Disclosure |
| | Standardization | Non-disclosure | Disclosure |
| Reliability | Third-party audits or evaluations | Non-disclosure | Disclosure |
| | Whether to refer to GRI | Non-disclosure | Disclosure |
| Completeness | Shareholder interests | Text qualitative description | Number quantitative description |
| | Creditor interests | Text qualitative description | Number quantitative description |
| | Employee interests | Text qualitative description | Number quantitative description |
| | Supplier interests | Text qualitative description | Number quantitative description |
| | Customer interests | Text qualitative description | Number quantitative description |
| | Environmental responsibility | Text qualitative description | Number quantitative description |
| | Financial responsibility | Text qualitative description | Number quantitative description |
| | System building and corporate governance | Text qualitative description | Number quantitative description |
| | Safety | Text qualitative description | Number quantitative description |
| | Public activities | Text qualitative description | Number quantitative description |
| | Sustainable development | Text qualitative description | Number quantitative description |
| | International exchange | Text qualitative description | Number quantitative description |
| | Description of the defect | Text qualitative description | Number quantitative description |

aspects, including readability, reliability and completeness of corporate sustainability information (due to the limited sample of sustainability reports disclosed by Chinese companies, this paper uses the ESG report or social responsibility report of the undisclosed sample company as an alternative). The measurement indicator system scores the symbolic disclosure behavior and substantive disclosure behavior of one company, and the specific measurement indicators are shown in Table 2. Consistent with many steps using content analysis, in the scoring process, each sample is scored by two people, and the two raters do not start the formal scoring until the consistency of the trial evaluation stage reaches more than 90%, and the difference between the two ratings is coordinated by a third person.

This paper uses the ratio of symbolic disclosure behavior to substantive disclosure behavior in sustainability to measure the level of "greenwashing" of corporate sustainability information. The higher the ratio, the higher the level of "greenwashing" of sustainability information; Conversely, the lower. The specific calculation formula of the "greenwashing" indicator of corporate sustainability information is shown below.

$$\mathrm{GI} = \frac{\sum_{i=1}^{17} X_i}{\sum_{i=1}^{17} Y_i + 1} \tag{5}$$

Among them, $X_i$ indicates the symbolic disclosure behavior of the corporate, and $Y_i$ indicates the substantive disclosure behavior of the corporate. If there is symbolic disclosure behavior, the $X_i$ is 1, and if there is substantive disclosure behavior, the $Y_i$ is 1. However, considering that some corporate may only have symbolic disclosure behaviors but no substantive disclosure behaviors, we add 1 to the denominator $Y_i$.

The mediating variables in this paper is degree of information asymmetry and quality of disclosure. Referring to previous literature [45–48], this paper constructs a comprehensive transparency indicator from the perspective of earnings quality, information disclosure evaluation index, number of analysts followed, analyst earnings forecast accuracy, and auditor to

**Table 3. The variable definition table used in this article.**

| Variable name | Variable type | Data source | Variable definition |
|---|---|---|---|
| CSV | Dependent | CSMAR | Determined according to the entropy weight method above |
| GI | Independent | / | Determined according to the above introduction |
| ILL | Mediation | / | Determined according to the above introduction |
| IDQ | Mediation | Wind | Determined according to the above introduction |
| SIZE | Control | CSMAR | The natural logarithm of the total asset |
| AGE | Control | CSMAR | Time to market |
| LEV | Control | CSMAR | Total Liabilities / Total Assets× 100% |
| ROA | Control | CSMAR | Net profit/total assets× 100% |
| GROWTH | Control | CSMAR | current year's revenue growth / previous year's total revenue ×100% |
| TAT | Control | CSMAR | Operating Income/Total Assets× 100% |
| TOP1 | Control | CSMAR | The shareholding ratio of the largest shareholder |
| BZ | Control | CSMAR | Number of board members |
| SZ | Control | CSMAR | Number of Supervisory Boards |
| IDR | Control | CSMAR | Number of Independent Directors / Number of Board of Directors× 100% |
| MARKET | Control | / | Fan Gang's marketization index |
| GDP | Control | CSMAR | The natural logarithm of GDP per capita |

measure the company's information transparency as a substitute indicator of information asymmetry (ILL).

As a signal screening and search mechanism, the ESG performance of companies can affect the quality of corporate information disclosure to a certain extent [49]. As an important embodiment of the concept of sustainability at the corporate level, investors and other stakeholders will identify the good or bad of the corporate based on the ESG score by third-party rating agencies. Studies have found that ESG disclosure helps companies access financing, and these investors tend to believe that companies with good ESG performance are better able to create competitive advantage and long-term value [50]. Therefore, this paper uses the Sino-Securities Index of ESG score as an alternative indicator of information disclosure quality (IDQ).

The control variables in this paper include two aspects. One is company level variables, including enterprise size, age, asset-liability ratio, total asset net interest rate, operating profit growth rate, total asset turnover rate, equity concentration, board size, supervisory board size, and proportion of independent directors; The other aspect is market level variables, including the degree of marketization of regions and the level of regional economic development. Table 3 shows the specific definitions of each variable.

## Data sources

This paper comprised companies that continuously publish sustainable development reports (or non-financial reports such as ESG reports and social responsibility reports) among China's A-share from 2010 to the end of 2021, which were selected based on the following criteria by a systematic deletion method: Considering that the information disclosed by financial and insurance companies is not comparable with other industries, it is excluded; Exclude ST and *ST during observation; Exclude companies with negative net assets; Eliminate companies with incomplete data.

According to our evaluations and by imposing the above limitations on the available population, a total of 8304 sample observations from 1241 companies were selected. The required

data comes from sustainability reports, ESG reports, social responsibility reports, CSMAR database, Wind, annual reports of listed companies, and CNI database. The statistical analysis of the data was done using Stata 16 and Excel software.

## Results and discussion

### Descriptive statistics

First, descriptive statistics analysis results are shown in Table 4. It is clear that the mean value of "shared value" creation (CSV) is 0.0399, the standard deviation is 0.0447, the minimum value is 0, and the maximum value is 0.955, which means that the "shared value" creation behavior between companies is uneven. The mean value of "greenwashing" (GI) of sustainability reporting is 5.705, the standard deviation is 4.270, the minimum value is 0.0769, and the maximum value is 13, which indicates that the overall level of "greenwashing" is high, and the degree of "greenwashing" varies greatly between different enterprises. The descriptive statistical results for the control variables are within the normal range.

### Correlation test

In order to preliminarily test the hypothesis presented above, Pearson correlation test is performed on the variables involved in the model. As can be seen in Table 5, with a correlation coefficient of -0.152 for "shared value" creation (CSV) and "greenwashing" (GI) and a significant negative at the 1% level, consistent with expectations, tentatively supporting the assumption of Hypothesis 1, and the control variables all significantly correlated with "shared value" creation (CSV). The correlation coefficient between the explanatory and control variables in the model was less than 0.5, and the variance inflation factor (VIF) was less than 2.5, indicating that the model did not have serious multicollinearity problems.

### Benchmark analysis

Before conducting the benchmark analysis, the industry effect and the province effect were controlled for considering that companies in different industries and provinces may face different risks. At the same time, Hausman test has a p-value of less than 0.05, rejecting the

**Table 4. Descriptive statistics.**

| VARIABLES | N | Mean | Standard Deviation | Median | Minimum | Maximum |
|---|---|---|---|---|---|---|
| CSV | 8304 | 0.0399 | 0.0447 | 0.0447 | 0 | 0.955 |
| GI | 8304 | 5.705 | 4.270 | 4.270 | 0.0769 | 13 |
| SIZE | 8304 | 22.30 | 1.360 | 1.360 | 18.38 | 28.64 |
| ROA | 8304 | 0.0510 | 0.0692 | 0.0692 | -2.646 | 0.604 |
| LEV | 8304 | 0.392 | 0.201 | 0.201 | 0.00708 | 1.269 |
| GROWTH | 8304 | -0.0810 | 33.41 | 33.41 | -2.353 | 1.261 |
| TAT | 8304 | 0.662 | 0.481 | 0.481 | 0.00404 | 8.787 |
| TOP1 | 8304 | 35.50 | 15.13 | 15.13 | 2.197 | 89.99 |
| BZ | 8304 | 8.756 | 1.755 | 1.755 | 3 | 18 |
| IDR | 8304 | 37.06 | 5.187 | 5.187 | 22.22 | 66.67 |
| SZ | 8304 | 3.696 | 1.213 | 1.213 | 1 | 11 |
| AGE | 8304 | 8.504 | 0.665 | 0.665 | 5.659 | 9.322 |
| MARKET | 8304 | 8.096 | 2.232 | 2.232 | -1.770 | 12.63 |
| GDP | 8304 | 10.93 | 0.285 | 0.285 | 10.34 | 11.30 |

**Table 5. Correlation coefficients.**

|  | CSV | GI | SIZE | ROA | LEV | TAT | TOP1 | BZ | IDR | SZ | AGE |
|---|---|---|---|---|---|---|---|---|---|---|---|
| CSV | 1 | | | | | | | | | | |
| GI | -0.152*** | 1 | | | | | | | | | |
| SIZE | 0.297*** | -0.379*** | 1 | | | | | | | | |
| ROA | -0.107*** | -0.013 | -0.058*** | 1 | | | | | | | |
| LEV | 0.244*** | -0.132*** | 0.511*** | -0.374*** | 1 | | | | | | |
| TAT | 0.037*** | -0.077*** | 0.006 | 0.086*** | 0.072*** | 1 | | | | | |
| TOP1 | 0.117*** | -0.076*** | 0.288*** | 0.062*** | 0.092*** | 0.068*** | 1 | | | | |
| BZ | 0.163*** | -0.113*** | 0.334*** | -0.038*** | 0.215*** | 0.0120 | 0.064*** | 1 | | | |
| IDR | -0.039*** | 0 | 0.004 | -0.012 | 0.00100 | -0.005 | 0.048*** | -0.442*** | 1 | | |
| SZ | 0.171*** | -0.107*** | 0.369*** | -0.065*** | 0.285*** | 0.041*** | 0.152*** | 0.406*** | -0.061*** | 1 | |
| AGE | 0.230*** | 0.045*** | 0.342*** | -0.177*** | 0.345*** | 0.070*** | 0 | 0.200*** | -0.034*** | 0.284*** | 1 |

Note:

*, **, and ***indicate, respectively, a significance level of 10%, 5%, and 1%.

$H_0$ hypothesis and a fixed-effect model was applied to test the Hypothesis 1. The results are shown in Table 6.

According to the regression results of the fixed-effect model in column (1), it can be seen that $R^2$ is 0.244, indicating that the model has a good good fit; CSV and GI are negatively correlated at the significance level of 1%, corresponding to the Hypothesis 1 that "greenwashing" behavior of information disclosure in sustainability reporting is negatively correlated with "shared value" creation. The results show that companies which "greenwash" sustainability information will seriously damage the "shared value" of stakeholders, and in turn bring unpredictable harm and loss to the entire market economy and society. This is consistent with the findings of Walker and Wan [4], on the one hand, corporate "greenwashing" behavior has a negative impact on its financial performance, and investors as "rational people" will reduce the losses caused by the negative financial performance of companies by selling stocks, etc., resulting in negative market reactions; On the other hand, the increase in environmental awareness has led to shareholders paying more and more attention to the companies' environmental performance. Marquis et al. [51], pointed out that corporate investors will evaluate their environmental performance through the environmental information disclosure, and will respond quickly once they find "greenwashing", which seriously affect the value sharing of the capital market.

## Robustness test

**Replace the explanatory variable.** The independent variable reflects the economic value in terms of the Tobin Q above, and theoretically, the higher the Tobin Q, the greater the economic value. However, economic value may be affected by other factors, and academia also has historical cost method, yield method, market value method, etc. for the calculation of economic value. To avoid the possible spurious regression caused by specific sample selected, this paper refers to the research of scholars [52], selects the return on assets (ROA) to reflect the economic value, and perform the regression on Model (1) again. The results are shown in column (2) of Table 6, which is consistent with the previous results.

**Lag the dependent variable.** This paper mainly studies the impact of "greenwashing" in sustainability reporting on the "shared value", but there are also companies that create more value that perform better in terms of sustainable development and are reluctant to "greenwash". In order to avoid errors caused by reverse causality, this paper tries to make the

**Table 6. Regression results of Sustainability reporting "greenwashing" and "shared value" creation.**

| VARIABLES | (1) | (2) | (3) | (4) | (5) |
|---|---|---|---|---|---|
| | CSV | CSV | CSV | CSV | CSV |
| GI | -0.001*** | -0.001*** | -0.001*** | -0.001*** | -0.001*** |
| | (-9.539) | (-10.044) | (-9.673) | (-6.172) | (-4.005) |
| SIZE | 0.006*** | 0.005*** | 0.006*** | 0.007*** | 0.006*** |
| | (12.159) | (19.279) | (11.801) | (11.298) | (4.503) |
| ROA | -0.039*** | -0.000 | -0.036*** | -0.037*** | -0.039** |
| | (-5.531) | (-0.227) | (-4.861) | (-4.681) | (-2.800) |
| LEV | 0.006** | 0.005*** | 0.005 | 0.004 | 0.006 |
| | (2.107) | (4.558) | (1.597) | (1.250) | (1.039) |
| GROWTH | 0.000 | -0.000 | 0.000 | 0.000 | 0.000 |
| | (0.325) | (-0.175) | (0.333) | (0.073) | (0.353) |
| TAT | -0.000 | -0.002*** | -0.000 | -0.000 | -0.000 |
| | (-0.421) | (-3.544) | (-0.004) | (-0.253) | (-0.197) |
| TOP1 | 0.000 | 0.000*** | 0.000 | 0.000* | 0.000 |
| | (1.464) | (3.062) | (1.486) | (1.725) | (0.915) |
| BZ | 0.000 | 0.000 | -0.000 | -0.000 | 0.000 |
| | (0.109) | (0.115) | (-0.503) | (-0.035) | (0.066) |
| IDR | -0.000 | -0.000 | -0.000 | -0.000 | -0.000 |
| | (-1.127) | (-0.769) | (-1.129) | (-0.710) | (-0.739) |
| SZ | -0.000 | 0.001*** | -0.000 | -0.001 | -0.000 |
| | (-0.108) | (3.726) | (-0.943) | (-1.048) | (-0.062) |
| AGE | 0.004*** | 0.004*** | 0.005*** | 0.003** | 0.004** |
| | (5.194) | (9.146) | (4.390) | (2.528) | (2.328) |
| MARKET | -0.001 | -0.001 | -0.001 | -0.002 | -0.001 |
| | (-0.930) | (-1.184) | (-0.608) | (-1.330) | (-0.480) |
| GDP | -0.032*** | -0.018*** | -0.038*** | -0.043*** | -0.032* |
| | (-9.998) | (-9.120) | (-10.148) | (-9.915) | (-1.869) |
| Constant | 0.234*** | 0.088*** | 0.296*** | 0.360*** | 0.234 |
| | (7.370) | (4.592) | (7.872) | (8.312) | (1.060) |
| Observations | 8304 | 8304 | 7226 | 6328 | 8304 |
| R-squared | 0.244 | 0.232 | 0.246 | 0.233 | 0.244 |

Note: The numbers in bracket is T value; *, **, and ***indicate, respectively, a significance level of 10%, 5%, and 1%.

dependent variable lag one period (L.GI), lag two period (L2.GI), and perform the regression on Model (1) again to test the hypothesis 1, and the results are shown in columns (3) and (4) of Table 6, which are consistent with the previous results.

**Individual and temporal double cluster regression.** To further solve the heteroscedasticity and autocorrelation problems of the sample, individual and temporal double cluster regression was used, and the results are shown in column (5) of Table 6, which is consistent with the previous results.

**Tool variable regression.** This paper makes the dependent variable (GI) lag one period (L.GI) and lag two period (L2. GI) as an instrumental variable for two-stage least squares regression in order to further alleviate the endogeneity problem caused by missing variables [53]. Table 7 shows the results of the instrumental variable regression, and the parameter estimates for the L.GI and L2.GI are 0.5561 and 0.2192, respectively, which are significant at the 1% level, indicating that the instrumental variable is significantly positively correlated with the

**Table 7. Tool variable regression results.**

| VARIABLES | (1) | (2) |
|---|---|---|
| | GI | CSV |
| L.GI | 0.5561*** | |
| | (0.006) | |
| L2.GI | 0.2192*** | |
| | (0.007) | |
| GI | | -0.0029*** |
| | | (0.000) |
| SIZE | -0.2542*** | 0.0031*** |
| | (0.019) | (0.000) |
| ROA | -0.0148 | -0.0002 |
| | (0.025) | (0.000) |
| LEV | 0.2759*** | 0.0069*** |
| | (0.101) | (0.002) |
| GROWTH | 0.0010 | -0.0000 |
| | (0.001) | (0.000) |
| TAT | -0.1367*** | -0.0013** |
| | (0.035) | (0.001) |
| TOP1 | -0.0027** | 0.0001*** |
| | (0.001) | (0.000) |
| BZ | -0.0365*** | -0.0001 |
| | (0.014) | (0.000) |
| IDR | 0.0053 | -0.0002*** |
| | (0.004) | (0.000) |
| SZ | -0.0274 | 0.0012*** |
| | (0.019) | (0.000) |
| AGE | 0.1493*** | 0.0022*** |
| | (0.041) | (0.001) |
| Market | 0.0488*** | -0.0025*** |
| | (0.010) | (0.000) |
| GDP | -1.2818*** | -0.0295*** |
| | (0.098) | (0.002) |
| KP-LM | | 7576.09*** |
| Cragg-Donald Wald F | | 12655.10 |
| Hansen J | | 0.355 |
| Constant | 19.5197*** | 0.3088*** |
| | (1.225) | (0.021) |
| Observations | 6204 | 6190 |
| R-squared | 0.619 | 0.097 |

Note: The numbers in bracket is T value; *, **, and ***indicate, respectively, a significance level of 10%, 5%, and 1%.

dependent variable. At the same time, the estimation results of instrumental variables need to meet a series of tests, such as the results of the unidentifiable test (KP-LM statistic) and weak instrumental variables (Cragg-Donald Wald F statistic) in Table 7, which show that the instrumental variables used do not have the problem of unrecognized and weak instrumental variables. Due to the use of two instrumental variables, an over-identification test is required, and the results of the over-identification test (Hansen J statistic) also show that there is no over-

identification problem. In summary, the tool variables used are reasonable and meet the requirements of effective tool variables. In the second stage of regression, the parameter estimates of GI were still significantly negative, indicating that the negative correlation between "greenwashing" behavior in corporate sustainability reporting and "shared value" creation was still valid after considering endogenous issues.

## Mediating effect test

For the mediation effect test of the degree of asymmetry of the whole sample, the results are shown in Table 8. Column (1) examines the overall effect of "greenwashing" of sustainability

**Table 8. Mediation effect test results.**

| VARIABLES | (1) | (2) | (3) | (4) | (5) | (6) |
|---|---|---|---|---|---|---|
|  | CSV | ILL | CSV | CSV | IDQ | CSV |
| GI | -0.001*** | -0.002** | -0.001*** | -0.001*** | -0.059*** | -0.001*** |
|  | (-2.658) | (-2.365) | (-2.732) | (-5.893) | (-11.537) | (-6.493) |
| ILL |  |  | -0.019* |  |  |  |
|  |  |  | (-1.863) |  |  |  |
| IDQ |  |  |  |  |  | -0.001* |
|  |  |  |  |  |  | (-1.900) |
| SIZE | 0.010*** | 0.052*** | 0.011*** | 0.006*** | 0.178*** | 0.006*** |
|  | (4.728) | (14.407) | (5.299) | (6.592) | (7.089) | (6.754) |
| ROA | -0.011 | 0.765*** | 0.003 | -0.040*** | 1.028*** | -0.039*** |
|  | (-0.652) | (16.680) | (0.146) | (-4.611) | (3.962) | (-4.486) |
| LEV | -0.006 | -0.080*** | -0.008 | 0.006 | -0.555*** | 0.005 |
|  | (-0.549) | (-4.086) | (-0.743) | (1.101) | (-4.176) | (0.960) |
| GROWTH | -0.000 | 0.000 | -0.000 | 0.000 | 0.000 | 0.000 |
|  | (-1.509) | (0.724) | (-1.270) | (0.405) | (0.012) | (0.429) |
| TAT | 0.001 | -0.004 | 0.000 | -0.001 | 0.062 | -0.001 |
|  | (0.278) | (-0.795) | (0.230) | (-0.376) | (1.417) | (-0.332) |
| TOP1 | -0.000 | 0.001** | -0.000 | 0.000 | 0.003 | 0.000 |
|  | (-1.154) | (2.428) | (-1.019) | (0.806) | (1.625) | (0.865) |
| BZ | 0.000 | 0.002 | 0.000 | -0.000 | 0.028** | 0.000 |
|  | (0.044) | (0.872) | (0.090) | (-0.005) | (1.988) | (0.058) |
| IDR | -0.000 | -0.000 | -0.000 | -0.000 | 0.001 | -0.000 |
|  | (-0.383) | (-0.020) | (-0.406) | (-0.716) | (0.193) | (-0.707) |
| SZ | 0.003* | 0.007*** | 0.003* | -0.000 | 0.038* | 0.000 |
|  | (1.734) | (2.609) | (1.847) | (-0.048) | (1.910) | (0.016) |
| AGE | 0.004 | -0.049*** | 0.003 | 0.004*** | 0.094** | 0.005*** |
|  | (0.638) | (-4.663) | (0.516) | (3.462) | (2.526) | (3.559) |
| MARKET | -0.000 | 0.009*** | -0.000 | -0.001 | 0.034 | -0.001 |
|  | (-0.136) | (5.844) | (-0.165) | (-0.636) | (1.106) | (-0.598) |
| GDP | -0.007 | -0.041*** | -0.007 | -0.032*** | -0.354*** | -0.033*** |
|  | (-0.791) | (-2.789) | (-0.789) | (-8.316) | (-3.359) | (-8.374) |
| Constant | -0.121 | -0.043 | -0.132 | 0.232*** | 5.236*** | 0.239*** |
|  | (-1.122) | (-0.217) | (-1.219) | (5.769) | (4.758) | (5.892) |
| Observations | 6399 | 6399 | 6399 | 8130 | 8130 | 8130 |
| R-squared | 0.324 | 0.288 | 0.327 | 0.244 | 0.236 | 0.244 |

Note: The numbers in bracket is T value; cluster standard error; *, **, and ***indicate, respectively, a significance level of 10%, 5%, and 1%.

reporting on "shared value" creation, and the coefficient of GI in the regression results is -0.001, which is significantly negative at the level of 1%, indicating that the "greenwashing" behavior significantly reduces "shared value" creation. Column (2) examines the influence of "greenwashing" of sustainability reporting on information asymmetry, and the coefficient of GI in the regression results is -0.002, which is significantly negative at the level of 1%, indicating that the "greenwashing" behavior significantly increases the degree of information asymmetry. In column (3), the coefficients of GI and ILL were both significant at the level of 1%, which indicates that the "greenwashing" behavior can indeed reduce "shared value" creation by affecting the degree of information asymmetry, and the degree of information asymmetry plays a partial mediation role, so as to verify the Hypothesis 2. In addition, the Bootstrap (95%) confidence interval P-value range from 0.0000141–0.0001185 (excluding 0), which further supports the mediation effect of information asymmetry.

For the mediation effect test of the quality of information disclosure of the whole sample, the results are shown in Table 8. Column (4) tests the overall effect of "greenwashing" of sustainability reporting on "shared value" creation, and the coefficient of GI in the regression results is -0.001, which is significantly negative at the level of 1%, indicating that the "greenwashing" behavior significantly reduces "shared value" creation. Column (5) tests the influence of "greenwashing" of sustainability reporting on the quality of information disclosure, and the coefficient of GI in the regression results is -0.059, which is significantly negative at the level of 1%, indicating that the "greenwashing" behavior significantly reduces the quality of information disclosure. The coefficients of GI and IDQ in column (6) were significant at the level of 1%, which indicates that the "greenwashing" behavior can indeed reduce "shared value" creation by affecting the quality of information disclosure, and the quality of information disclosure plays a partial mediation role, so as to verify Hypothesis 3. In addition, the P-value of the confidence interval of Bootstrap (95%) is 0.0000607–0.000165 (excluding 0), which further supports the mediation effect of the quality of information disclosure.

## Further analysis

The above analysis results show that the "greenwashing" behavior of corporate sustainability reporting will affect the "shared value" by increasing the degree of information asymmetry and reducing the quality of information disclosure, which means that the problem of "greenwashing" is imminent. At present, although economic globalization is developing, environmental pollution, waste of resources and other ecological problems are becoming more and more serious. Therefore, sustainability has received more and more attention from all countries in the world. In order to avoid corporate misconduct which, threaten the harmonious and sustainability of the entire market economy, it is increasingly important to explore effective governance strategies for the "greenwashing" of information disclosure in sustainability reporting. The following is discussion of "filtering" governance measures from the perspectives of different internal and external stakeholders.

**Internal "filtration": Internal control level and "greenwashing".**   Imperfect internal control over information disclosure may make the "greenwashing" of companies unimpeded. Without internal controls that check and balance each other, financial reporting is bound to be of poor quality and fraudulent [54, 55]. This is true for financial reporting, so is sustainability reporting, and even more. Therefore, this paper selects the internal control index (ICC) of Dibo database, takes "shared value" as the dependent variable, and uses the interaction item of "greenwashing" of sustainability reporting and internal control as independent variables for regression analysis, and the results are shown in Table 9. The GI×ICC coefficient is positive and significant at the level of 5%, indicating that when the quality of internal control is high,

**Table 9. Further analysis results.**

| VARIABLES | (1) | (2) | (3) | (4) |
|---|---|---|---|---|
| | CSV | CSV | CSV | CSV |
| GI | -0.001*** | -0.003*** | -0.001*** | -0.001*** |
| | (-3.649) | (-3.989) | (-3.752) | (-3.588) |
| ICC | -0.025*** | | | |
| | (-3.113) | | | |
| GI×ICC | | 0.002** | | |
| | | (2.332) | | |
| JF | | | -0.000** | |
| | | | (-3.054) | |
| GI×JF | | | | 0.000** |
| | | | | (3.153) |
| SIZE | 0.006*** | 0.007*** | 0.008*** | 0.008*** |
| | (4.680) | (5.108) | (7.184) | (7.567) |
| ROA | -0.044** | -0.039* | -0.026** | -0.028** |
| | (-2.433) | (-2.178) | (-2.384) | (-2.526) |
| LEV | 0.005 | 0.005 | 0.010 | 0.008 |
| | (0.824) | (0.689) | (1.479) | (1.284) |
| GROWTH | 0.000 | 0.000 | 0.000 | 0.000 |
| | (0.356) | (0.380) | (0.354) | (0.291) |
| TAT | -0.001 | -0.000 | 0.002 | 0.002 |
| | (-0.243) | (-0.064) | (0.884) | (1.009) |
| TOP1 | 0.000 | 0.000 | 0.000 | 0.000 |
| | (0.920) | (0.998) | (0.716) | (0.798) |
| BZ | 0.000 | 0.000 | 0.000 | 0.000 |
| | (0.031) | (0.061) | (0.060) | (0.041) |
| IDR | -0.000 | -0.000 | -0.000 | -0.000 |
| | (-0.562) | (-0.544) | (-1.148) | (-1.135) |
| SZ | -0.000 | -0.000 | 0.000 | 0.000 |
| | (-0.223) | (-0.134) | (0.106) | (0.219) |
| AGE | 0.005** | 0.005** | 0.004* | 0.004 |
| | (2.619) | (2.458) | (1.883) | (1.735) |
| MARKET | -0.001 | -0.001 | 0.001 | 0.001 |
| | (-0.595) | (-0.584) | (1.005) | (1.020) |
| GDP | -0.032 | -0.034* | -0.013*** | -0.014*** |
| | (-1.777) | (-1.825) | (-3.261) | (-3.661) |
| Constant | 0.226 | 0.251 | -0.019 | -0.014 |
| | (0.981) | (1.079) | (-0.414) | (-0.300) |
| Observations | 7628 | 7628 | 7196 | 7196 |
| R-squared | 0.243 | 0.245 | 0.270 | 0.272 |

Note: The numbers in bracket is T value; cluster standard error; *, **, and ***indicate, respectively, a significance level of 10%, 5%, and 1%.

the negative impact of "greenwashing" on "shared value" creation in the sustainability reporting will be weakened.

**External "filtering": Media supervision and "greenwashing".** The problem of "greenwashing" is ultimately a problem of information asymmetry, and there is a difference in the understanding of "greenwashing" about the real information between companies and

audiences. "Greenwashing" companies with sufficient private information can establish a good environmental image and obtain financing facilities through the regulation of their behaviors [27]. Therefore, it is necessary to increase the supply of information to facilitate the display of the truth in the face of information asymmetry. And through third-party mechanisms such as media supervision, it is helpful to improve stakeholders' awareness of the current situation of sustainability of companies and improve information transparency.

This paper refers to the previous research [56–58], uses the number of positive reports, negative reports, and neutral reports provided by the financial database of China Research Data Service Platform (CNRDS), and designs the Janis-Fadner coefficient (JF) to construct media supervision indicators, as shown below.

$$f(x) = \begin{cases} \dfrac{e^2 - ec}{t^2}, & e > c \\ \dfrac{ec - e^2}{t^2}, & e > c \\ 0, & e = c \end{cases} \tag{6}$$

where e is the number of positive media reports, c is the number of negative media reports, and t is the sum of the number of positive reports and the number of negative reports. The JF coefficient can range from (-1,1). When there are more positive reports about the company, the closer the JF coefficient is to 1, and the less pressure the company faces on media supervision. When there are more negative reports about the company, the JF coefficient is close to -1, and the company is under greater pressure on media supervision. Taking "shared value" creation as the dependent variable, the interaction item of "greenwashing" of sustainability reporting and media supervision as independent variables for regression analysis, we can find the results in Table 9. The GI×JF coefficient is positive and significant at the level of 5%, indicating that when companies are under greater pressure from external media supervision, the negative impact of "greenwashing" on "shared value" creation in sustainability reporting will be weakened. Lokuwaduge and Heenetigala [59], examined the influencing factors of social and governance reporting and found that companies' motivation to voluntarily disclose social and governance reporting was highly influenced by regulatory isomorphic factors, so "greenwashing" behavior could naturally be curbed when external regulatory pressure increased.

## Conclusions

Guided by the concept of sustainability, this paper discusses the relationship between the "greenwashing" behavior of information disclosure in sustainability reporting and the "shared value" creation. The results show that the "greenwashing" behavior has a significant negative impact on the creation of "shared value", that is, the occurrence of "greenwashing" behavior significantly reduces the level of "shared value" creation of companies, and the results remain unchanged after a series of robustness and endogenous tests. The results of the mediation effect test show that the degree of information asymmetry and the quality of information disclosure play a partial mediation role between them, which indicates that the "greenwashing" behavior can reduce "shared value" creation by affecting the transmission of information effect. Further analysis shows that the more sound the internal control of companies and the greater the pressure of external media supervision, the more conducive it is to weaken the negative impact of "greenwashing" on "shared value" creation.

The policy implications brought by the above conclusions to encourage companies to achieve the goal of "shared value" creation and further enhance sustainable development capabilities include the following:

First, promote legislative work, formulate unified standards, and compress the "greenwashing" gray space. The quality of sustainability reporting disclosed by different countries and even different companies in the same country is not uniform, and there is "greenwashing" to varying degrees. Therefore, through legislation, standardizing the compilation and disclosure of sustainability reporting, we can fundamentally rectify and curb "greenwashing" behavior, improve the quality of sustainable development information, and escort carbon peaking and carbon neutrality.

Second, implement mandatory disclosure, implement independent assurance, and strengthen "greenwashing" social supervision. At this stage, sustainability reporting is mainly voluntary disclosure, voluntary disclosure lacks rigid constraints, it is easy to breed selective disclosure and "greenwashing" behavior of reporting only the good piece of news and hiding the bad one. Since "greenwashing" can bring huge economic benefits, in the face of this interest temptation, it is obviously unrealistic to expect companies and financial institutions to consciously and voluntarily suppress the "greenwashing" impulse. Implementing a mandatory disclosure system for sustainability reporting and introducing an independent assurance mechanism to improve the transparency of sustainability information disclosure and to increase the disclosure obligations and responsibilities of enterprises are essential. This can give external stakeholders such as the public and the news media the opportunity to strengthen the supervision of enterprise "greenwashing" behavior.

Third, with the help of digital empowerment, strengthen capacity-building and improve the "greenwashing" governance mechanism. The development of digital technologies such as artificial intelligence, blockchain, cloud computing, big data and the Internet of Things enables us to establish a powerful information system to systematically collect, efficiently analyze and accurately trace sustainable development-related information. They can also help managers formulate and implement sustainable development-related strategies and risk management, enhance the capacity building of managers, improve the governance mechanism of sustainability, and provide solid basic data and information for the sustainable development of companies and society.

There are still some shortcomings in this study. First, mainly due to the non-mandatory disclosure of sustainability reporting, the sample size of sustainability information obtained is limited, so the research conclusions of this paper may be affected by the measurement error of "greenwashing" of sustainability information. Therefore, the study subjects need to be expanded in the future. Second, the use of the Sino-Securities Index of ESG score as a substitute variable for the quality of sustainability information disclosure in this paper may also have measurement errors, which will affect the scientific validity of the paper's empirical conclusions. It is necessary to explore other more appropriate variables in future studies.

## Supporting information

**S1 Table. Sustainability reporting "greenwashing" indicator system supplemental information.**
(DOCX)

**S2 Table. Correlation coefficients.**
(DOCX)

**S3 Table. VIF test results.**
(DOCX)

## Author Contributions

**Conceptualization:** Wei Xu.

**Data curation:** Mingzhu Li, Sen Xu.

**Formal analysis:** Wei Xu, Sen Xu.

**Funding acquisition:** Wei Xu.

**Investigation:** Mingzhu Li, Sen Xu.

**Methodology:** Wei Xu, Mingzhu Li, Sen Xu.

**Project administration:** Wei Xu.

**Resources:** Sen Xu.

**Software:** Sen Xu.

**Supervision:** Wei Xu, Sen Xu.

**Validation:** Wei Xu, Mingzhu Li, Sen Xu.

**Visualization:** Wei Xu, Mingzhu Li, Sen Xu.

**Writing – original draft:** Wei Xu, Mingzhu Li.

**Writing – review & editing:** Wei Xu, Sen Xu.

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
