## [Decision Letter · Decision Letter 0]

28 Nov 2022

PONE-D-22-30440Unveiling the "Veil" of information disclosure: Sustainability reporting "Greenwashing" and "Shared value"PLOS ONE

Dear Dr. Li,

Thank you for submitting your manuscript to PLOS ONE. After careful consideration, we feel that it has merit but does not fully meet PLOS ONE’s publication criteria as it currently stands. Therefore, we invite you to submit a revised version of the manuscript that addresses the points raised during the review process.

*Based on the comments of the two blind reviewers I have decided to accept your paper with MAJOR REVISIONS, at the bottom are the detailed review comments.*

We look forward to receiving your revised manuscript.

Kind regards,

Vincenzo Basile, PhD

Academic Editor

PLOS ONE

Journal Requirements:

Reviewers' comments:

Reviewer's Responses to Questions

**Comments to the Author**

1. Is the manuscript technically sound, and do the data support the conclusions?

Reviewer #1: Partly

Reviewer #2: Yes

2. Has the statistical analysis been performed appropriately and rigorously? 

Reviewer #1: Yes

Reviewer #2: No

3. Have the authors made all data underlying the findings in their manuscript fully available?

Reviewer #1: No

Reviewer #2: No

4. Is the manuscript presented in an intelligible fashion and written in standard English?

Reviewer #1: No

Reviewer #2: Yes

5. Review Comments to the Author

Reviewer #1: Follow the review report strictly and revise the paper, there are many discrepancies in the study. The author should deeply review the paper and revise the paper according to the points of review report.

Reviewer #2: This paper about information disclosure and sustainability reporting.

After carefully reading the paper I have the following comments to the authors:

1. The sample needs to described and relevant statistics such as mean, standard deviation, kurtosis and asymmetry needs to be displayed. A normality test should also be displayed.

2. The assumptions of the statistical model needs to be verified.

3. Residual tests should also be presented in a detailed manner.

After revised the paper should be resubmitted.

6. PLOS authors have the option to publish the peer review history of their article (what does this mean?). If published, this will include your full peer review and any attached files.

Reviewer #1: No

Reviewer #2: No

---

## [Author Response · Author response to Decision Letter 0]

13 Dec 2022

Dear Dr. Vincenzo Basile and Reviewers,

Thank you for your letter and for the reviewers’ comments concerning our manuscript entitled “Unveiling the ‘Veil’ of information disclosure: Sustainability reporting ‘Greenwashing’ and ‘Shared value’” (PONE-D-22-30440). Those comments are all valuable and very helpful for revising and improving our paper, as well as the important guiding significance to our researches. We have studied comments carefully and have made correction which we hope meet with approval. Revised portion are marked in red in the Revised Manuscript with Track Changes. The reviewers’ comments are laid out below in italicized font and specific concerns have been numbered. Our response is given in the blue text.

Responds to the reviewer’s comments:

1. Response to comment: Title is good try to make it more attractive.

Response: Thank you for the title suggested. Our manuscript entitled “Unveiling the ‘Veil’ of information disclosure: Sustainability reporting ‘Greenwashing’ and ‘Shared value’”, which fits well with the content of this research. As reviewer suggested that it is remarkable. So, after careful consideration, we decided not to change it for the time being and thank you very much for your valuable suggestions.

2. Response to comment: Abstract: The methods can be used, however it would be helpful to have more background information on why they were selected.

Response: We think this is an excellent suggestion. On the basis of the original, we modify the abstract and relevant content are added to the Revised Manuscript with Track Changes (Line 25-31): With the increasing attention of the capital market to corporate environmental, social and governance information, sustainability reporting has become an important carrier for stakeholders to gain insight into sustainability of companies. But the emerged “greenwashing” problem has also brought haze to the value creation among capital market. Its role is to study the consequences of the pseudo-social responsibility behavior of “greenwashing”.

3. Response to comment: The abstract would be fuller with policy suggestions included.

Response: It is really true as reviewer suggested that we lack policy suggestions in the abstract section, so we have added them in the Revised Manuscript with Track Changes (Line 48-53): The results call for the state to promote legislative work, formulate unified standards and compress the “greenwashing” gray space; Governments could implement mandatory disclosure, implement independent authentication and strengthen “greenwashing” social supervision; Companies should strengthen capacity building and improve the “greenwashing” governance mechanism with the help of digital empowerment.

4. Response to comment: This is a well-written paper with intriguing implications for learning more about the origins, progressions, and longevity of eating disorders, but its relevance to nursing is unclear.

Response: We agree with you that the content of our manuscript has limitations, its relevance to nursing is unclear. In fact, this is an exciting future area. Its main research purpose is to study the consequences of the pseudo-social responsibility behavior of “greenwashing”, therefore, more attention will be paid to the issue of corporate sustainability in the paper. In the Introduction and Literature Review section, we have gone to great lengths to explain the origins, progressions, and longevity of corporate sustainability “greenwashing” problem, and we have added theoretical and empirical context in the Revised Manuscript with Track Changes (Line 204-220) to supplement the lack of relevant theoretical explanations.

5. Response to comment: The author has to clarify and elaborate on why examining distinct dimensions of work engagement is useful.

Response: Thanks for your suggestion. As you can see, we have examined distinct dimensions of corporate values to explain “shared value”. In Literature Review section, we have combed through the research of relevant scholars and the sustainable development theory advocates the concept of creating “shared value” (Elkington, 1994; 1997). We have added more content in the Revised Manuscript with Track Changes (Line 157-159, Line 425-430) to clarify and elaborate: From the above, it can be seen that the sustainable development theory advocates the concept of creating “shared value”, and companies will help promote the value sharing with society and the environment by investing in social and environmental improvement activities that can enhance their competitiveness, and form a symbiotic relationship in which corporate success and social progress complement each other. In general, corporate “shared value” creation activities can create different types of value for stakeholders, namely economic, social and environmental (Verboven, 2011). Therefore, this paper uses this perspective to comprehensively measure the “shared value” creation activities of companies from three aspects: economic value, social value, and environmental value.

6. Response to comment: Newer studies demonstrating the value of having a supportive superior should be added.

Response: We sincerely appreciate this valuable comment. We have checked the literature carefully and added more references on having a supportive superior and the relevant factors of “greenwashing” and “shared value” in the Revised Manuscript with Track Changes (Line 204-220) to support this idea, and new added literatures have listed in the Revised Manuscript ([28, 29, 30, 31]).

7. Response to comment: Important normalcy assumptions should be clarified.

Response: We thank the reviewer of pointing out this issue. We indeed should have applied described and relevant statistics and more normality test to clarify its important assumptions. On this basis, first, we have added Descriptive Statistics and Correlation test in the Revised Manuscript with Track Changes (Line 519-540) to verify the assumptions of the statistical model, such as mean, Standard Deviation, Median, Minimum, Maximum, and the variance inflation factor (VIF). See for details in Table 4 and Table 5 in the Revised Manuscript. Second, the Hausman test has a p-value of less than 0.05, rejecting the H_0 hypothesis and a fixed-effect model was applied to test our assumptions in the Revised Manuscript with Track Changes (Line 542-546). Third, it can be seen that results of the fixed-effect model R^2 is 0.244, indicating that the model has a good good fit in the Revised Manuscript with Track Changes (Line 547-548). We hope our data could clearly verify the Empirical assumptions and results in the paper and be well displayed.

8. Response to comment: In the abstract, they should briefly discuss the most significant results of their research.

Response: It is really true as reviewer suggested that we lack a briefly discuss the most significant results of our study in the abstract, so we have re-written and summarize this part. In the Abstract section, we have added a discussion of the results after summarizing the findings, including the practical significance and comparisons with other existing studies in the Revised Manuscript with Track Changes (Line 41-45): This paper enriches the literature on the economic consequences of “greenwashing” in corporate sustainability disclosure and the influencing factors of “shared value” creation, extends the research on information disclosure and “shared value” from financial information to non-financial information. In the Introduction section, we have added a discussion of the questions and briefly discuss the most significant results of our research, which we think this will make the paper more informative in the Revised Manuscript with Track Changes (Line 102-123): This research attempts to answer the following questions: 1) Does the “greenwashing” behavior of corporate sustainability reporting and its information disclosure affect the “shared value” creation of enterprises? 2) If there is an influence between them, how does the influence path unfold? and 3) What kind of constraint mechanism can maximize the negative impact of the “greenwashing” behavior in corporate sustainability reporting? 

Accordingly, this paper comprehensively considers the “shared value” creation needs of environmental, social, enterprise and other stakeholders, and use the data of China’s listed companies for empirical testing. It is found that the “greenwashing” behavior of information disclosure in corporate sustainability reporting significantly reduces “shared value” creation, and the degree of sustainability information asymmetry and information disclosure quality play a partial role as an intermediary between them. Further analysis shows that the more sound the internal control of the enterprise and the greater the pressure of external media supervision, the more conducive it is to weaken the negative impact of “greenwashing” on “shared value” creation. The research conclusion enriches the relevant literature on the economic consequences of “greenwashing” in sustainability disclosure and the influencing factors of “shared value” creation, and extends the research of information disclosure and “shared value” from financial information to non-financial information, which provides scientific evidence for improving the quality of information disclosure in sustainability reporting, achieving the goal of “shared value” creation, and further improving corporate sustainable development capabilities.

9. Response to comment: The work has to detail its key contributions, such as new methods or insights.

Response: Based on the research results of previous scholars in the Literature Review section, we have complemented this part in the Revised Manuscript with Track Changes (Line 231-255): In summary, the existing research provides theoretical guidance and thoughtful reference for the study of this paper. However, the following issues need further research: First, the influencing factors of “shared value” creation currently mainly focused on the research of various factors such as innovation ability and big data on “shared value” creation, and the exploration of situational influencing factors such as “greenwashing” is insufficient and scattered. Second, since the information effect is subject to the quantity and quality of information disclosure, this paper distinguishes the information effect brought by the “greenwashing” behavior into the degree of information asymmetry (quantity) and the quality of information disclosure (quality), and explores the mediating role of information asymmetry and information disclosure quality respectively. Third, existing research mainly studies the governance of “greenwashing” behavior from the perspectives of policy system and macro collaborative governance, however, there are few studies on multi-level and integrated governance strategies that “filter” sustainability reporting. According to the accurate identification of “greenwashing” behavior of corporate sustainability information, this paper’s findings reveal the mechanism of “greenwashing” behavior and a company’s “shared value” creation. It also explores scientific and reasonable “filtering” governance strategies, which can extend the research on information disclosure and “shared value” from financial information to non-financial information. The findings could help to improve the quality of information disclosure in sustainable development reports, and encourages corporates companies to achieve the goal of “shared value” creation, and further enhance sustainable development capabilities.

10. Response to comment: Author should offer theoretical and empirical context for important variables. Adding solid theoretical explanations for underlying links between variables is recommended to strengthen this section. 

Response: In view of this valuable suggestion, we have revised the content of the paper and added more new references to clarify the underlying links between variables. And research on sustainability reporting "greenwashing" and “shared value” creation has been supplemented in the Revised Manuscript with Track Changes (Line 204-220): At present, the research data on the impact of “greenwashing” behavior on “shared value” is scarce, but “shared value” is actually a comprehensive concept, which can be reflected in the financial performance, environmental performance, and social behavior of companies to a certain extent. First, fulfilling social responsibility can help companies profit from capital markets, and in the long run, substantial environmental actions can improve corporate financial performance (Cai and He, 2014); Once the “greenwashing” activity is recognized by investors, the capital market will respond to it (Torelli et al., 2020), which is reflected in the more serious the “greenwashing” situation, the lower the cumulative excess yield during the window period. Secondly, “greenwashing” activities will reduce the enthusiasm of companies for green product research and development, reduce consumers' willingness to consume green products (Nguyen et al., 2019), mislead stakeholders to make decisions, and lead to a vicious circle of corporate environmental performance. Finally, “greenwashing” will imitate and spread within the group, forming a “ripple effect”, causing other companies to imitate (Huang and Zhao, 2018), distorting the ethical standards of the industry or region, and people will question the authenticity of the proposition of social responsibility, and eventually weaken the social responsibility awareness of the entire group.

11. Response to comment: It is suggested that the article be revised since the section on policy suggestions is very simple and the author did not provide thorough policy recommendations based on the study results presented here. In order to strengthen the discussion sections, it is helpful to draw parallels between the current study and others that have been conducted on similar topics.

Response: In order to take reviewer’s concern into account, and improve the quality of our manuscript, we have drawn parallels between the current study and others on similar topics. First, studies consistent with the conclusions of our Benchmark analysis’ results are discussed in the Revised Manuscript with Track Changes (Line 554-564): This is consistent with the findings of Walker and Wan et al.(2012), on the one hand, corporate “greenwashing” behavior has a negative impact on its financial performance, and investors as "rational people" will reduce the losses caused by the negative financial performance of companies by selling stocks, etc., resulting in negative market reactions; On the other hand, the increase in environmental awareness has led to shareholders paying more and more attention to the companies’ environmental performance. Marquis and Toffel (2016), pointed out that corporate investors will evaluate their environmental performance through the environmental information disclosure, and will respond quickly once they find “greenwashing”, which seriously affect the value sharing of the capital market. Second, studies consistent with the conclusions of our Extensive test’s results are discussed in the Revised Manuscript with Track Changes (Line 704-709): Lokuwaduge and Heenetigala (2017), examined the influencing factors of social and governance reporting and found that companies’ motivation to voluntarily disclose social and governance reporting was highly influenced by regulatory isomorphic factors, so “greenwashing” behavior could naturally be curbed when external regulatory pressure increased. It is hoped that the addition to the above discussion will help to strengthen this paper.

13. Response to comment: If you can, cut out some of the tangential details that aren't essential to the study's argument.

Response: Thank you very much for your kind consideration of this manuscript. We agree with this suggestion and have modified and cut out some of details that aren't essential to the study as much as possible. However, due to the content and structure of the paper, the content we cut may only be a minority, and we believe that every sentence in the discussion section exists to support our conclusions. We apologize for any inconvenience this may cause to reviewers.

14. Response to comment: Is the manuscript technically sound, and do the data support the conclusions?

Response: To study the consequences of the pseudo-social responsibility behavior of “greenwashing”, this paper takes China's listed companies as the research sample to empirically examine the relationship between sustainability reporting “greenwashing” and “shared value” creation. We have drawn our conclusions by proposing research hypotheses, descriptive statistics, correlation analysis, regression analysis, and robustness testing on experimental data. So, our manuscript has described a technically sound piece of scientific research with data that supports the conclusions. Moreover, the experiments in the paper have been conducted rigorously. In Methodology and data section, we show the variable design of the paper as well as the source of the data, with appropriate controls, replication, and sample sizes. By listing the above content, we present our experimental results in full.

15. Response to comment: Has the statistical analysis been performed appropriately and rigorously?

Response: Before conducting our experiment, we first performed the Hausmann test to prove a fixed-effect model should be applied to test our assumptions in the paper. Then we validated the research assumptions using regression analysis, and some robustness testing methods, such as replacing the explanatory variable, lagging the dependent variable, individual and temporal double cluster regression, and tool variable regression were used to further confirm the robustness of the benchmark regression results. However, we are very sorry for our negligence as reviewer suggested that 1. The sample needs to described and relevant statistics such as mean, standard deviation, kurtosis and asymmetry needs to be displayed. A normality test should also be displayed. 2. The assumptions of the statistical model needs to be verified. 3. Residual tests should also be presented in a detailed manner. Considering the reviewer’s suggestion, we have added Descriptive Statistics and Correlation test in the Revised Manuscript with Track Changes (Line 519-540) to verify the assumptions of the statistical model, such as mean, Standard Deviation, Median, Minimum, Maximum, and the variance inflation factor (VIF). See for details in Table 4 and Table 5 in the Revised Manuscript.

16. Response to comment: Have the authors made all data underlying the findings in their manuscript fully available?

Response: There are no restrictions on publicly sharing data in our research, but since most of the data in our paper is collated from third-party data from the CSMAR database, we did not disclose our data when we first submitted. Considering the requirements of journal editors and reviewers, we have uploaded the raw data used in our paper in the Supporting information (S_4Table) for review.

17. Response to comment: Is the manuscript presented in an intelligible fashion and written in standard English?

Response: We apologize for the poor language of our manuscript. We worked on the manuscript for a long time and repeated addition and removal of sentences and sections obviously led to poor readability. We have now worked on both language and readability and have also involved native English speakers for language corrections. We really hope that the flow and language level have been substantially improved.

We tried our best to improve the manuscript and made some changes in the manuscript. These changes will not influence the content and framework of the paper. And here we did not list the changes but marked in red in the Revised Manuscript with Track Changes.

We appreciate for your warm work earnestly, and hope that the correction will meet with approval.

Once again, thank you very much for your comments and suggestions.

Yours sincerely,

Mingzhu Li

Corresponding author

Business School, Xi’an University of Finance and Economics, Xi’an, Shaanxi, China

2121031016@xaufe.edu.cn

---

## [Editor Report · Decision Letter 1]

19 Dec 2022

Unveiling the "Veil" of information disclosure: Sustainability reporting "Greenwashing" and "Shared value"

PONE-D-22-30440R1

Dear Dr. Mingzhu Li,

We’re pleased to inform you that your manuscript has been judged scientifically suitable for publication and will be formally accepted for publication once it meets all outstanding technical requirements.

Kind regards,

Vincenzo Basile, PhD

Academic Editor

PLOS ONE
---

## [Editor Report · Acceptance letter]

22 Dec 2022

PONE-D-22-30440R1 

Unveiling the "Veil" of information disclosure: Sustainability reporting "greenwashing" and "shared value" 

Dear Dr. Li:

I'm pleased to inform you that your manuscript has been deemed suitable for publication in PLOS ONE. Congratulations! Your manuscript is now with our production department. 

Kind regards, 

on behalf of

Dr. Vincenzo Basile 

Academic Editor

PLOS ONE